# Fear, stress, anxiety, depression and insomnia related to COVID-19 among undergraduate nursing students: An international survey

**Mohammed Al Maqbali**[1☯], **Norah Madkhali**[2☯], **Alexander M. Gleason**[3☯], **Geoffrey L. Dickens**[4,5☯]*

**1** Fatima College of Health Sciences, Al Ain, United Arab Emirates, **2** Jazan University, Jizan, Saudi Arabia, **3** Fatima College of Health Sciences, Abu Dhabi, UAE, **4** Mental Health Nursing Department of Nursing, Midwifery and Health Faculty of Health and Life Sciences, Northumbria University, Newcastle-Upon-Tyne, United Kingdom, **5** Adjunct Professor Western Sydney University, Penrith, Australia

☯ These authors contributed equally to this work.
* geoffrey.dickens@northumbria.ac.uk

**Data Availability Statement:** As the data are collected from four countries, they are not publicly available due to the Rules Governing the Ethics of Scientific Research. Other researchers may obtain

## Abstract

The emergence of COVID-19 has produced unprecedented change in daily life activities leading to major impacts on psychological wellbeing and sleep among individuals worldwide. The study aimed to assess levels of fear, stress, anxiety, depression, and insomnia among undergraduate nursing students in four countries two years after the start of the pandemic. An international, multi-centre cross-sectional electronic survey was conducted between December 2021 and April 2022. An on-line questionnaire was distributed via Qualtrics® and JISC® software. Instruments included the Fear of COVID-19 Scale, the Perceived Stress Scale, the Hospital Anxiety and Depression Scale and the Insomnia Severity Index, and a demographics and academic background questionnaire. The independent variables included demographic and academic backgrounds, while fear level, stress, anxiety, depression, and insomnia were the dependent variables. A total of 918 undergraduate nursing students from KSA, Oman, UK, and UAE were participants in the study. Students presented with stress (91.6%), anxiety (69.1%), depression (59.8%), and insomnia (73.2%). The participants' mean Fear of COVID-19 Scale score was 12.97 (SD = 6.14). There were significant positive relationships between fear of COVID-19, stress, anxiety, depression, and insomnia. Undergraduate nursing students experienced moderate to severe levels of Fear of COVID-19, stress, anxiety, depression, and insomnia two years after the onset of the COVID-19 pandemic. Psychological intervention and peer support are needed to reduce the long-term adverse outcomes of mental health problems and insomnia. It is important to introduce education about crisis management of infectious disease during pandemics into the nursing curriculum to increase student knowledge and improve their preparedness for such emergencies.

access to the data directly from the Ethics Review Committee at Fatima College of Health Sciences (fchs.ethics@fchs.ac.ae).

**Funding:** The authors received no specific funding for this work.

**Competing interests:** The author(s) declared no potential conflicts of interest with respect to the research, authorship, and/or publication of this article.

# 1 Introduction

The world has changed considerably since December 2019 when the novel coronavirus disease (COVID-19) emerged in Wuhan City, Hubei province, China, and rapidly spread worldwide [1]. The World Health Organization (WHO) declared COVID-19 to be a pandemic on January 30, 2020 [2]. COVID-19 has posed a serious threat to human health, notably there have been more than 750 million fatalities associated with the disease [3]. Additionally, the uncertain nature of consequences of the disease has caused unexpected changes in social life, work, and travel activities.

Since the moment WHO declared COVID-19 as a pandemic, governments throughout the world implemented strict measures including lockdowns to reduce the transmission rate of the virus [4]. Consequently, higher education institutions were closed and most shifted to remote modes of educational delivery. Nursing students experienced a decrease in traditional classroom learning due to the pandemic, and at the same time, they were required to engage in essential clinical practice places as part of their education. This double impact affected their learning experience and demanded additional responsibilities during their training. Several researchers found that nursing students might experience of anxiety and stress during clinical placement [5, 6]. It is known that, even outside the context of a global pandemic, nursing students indicate that difficult learning materials, stringent examinations, long hours of study, the challenges of clinical placements, and the physical and emotional demands of programs can lead to mental health problems such as, stress, anxiety, and depression [7–9].

Previous studies during Severe Acute Respiratory Syndrome and Middle East Respiratory Syndrome outbreaks revealed that nursing students experienced high stress levels due to their increased risk of infection from direct patient contact [10–12]. Similarly, the mental health of pre-registration nursing students has been affected during the COVID-19 pandemic, as evidenced by reports of negative emotions, fear, confusion, pessimism, sleep disturbance, and an increasing number of psychological problems [13, 14]. This can lead to increased attrition rates from nursing education [15].

However, the mental health status of undergraduate nursing students after two years of the COVID-19 pandemic has yet to be explored. The contribution of this study, therefore, is to ascertain the mental health status of nursing students two years after the WHO first identified COVID-19 as a pandemic. This is crucial in enabling better planning for interventions to prevent and manage the mental health problems of nursing students in the event of the emergence of similar or other diseases in the future. The specific objectives of this study were to describe the prevalence of fear, stress, anxiety, depression, and insomnia, and their relationship with academic demographic variables, among undergraduate nursing students in Kingdom of Saudi Arabia (KSA), Oman, United Kingdom (UK), and United Arab Emirates (UAE) after more than two years of the COVID-19 pandemic.

# 2 Methods

## 2.1 Study design

This international collaborative study used a web-based cross-sectional design and was conducted in four countries (KSA, Oman, UK, and UAE). The sample consisted of undergraduate nursing students who were attending higher education institutions in participating countries (UK—Northumbria University, Oman—Oman College of Health Sciences, Saudi Arabia—Jazan University, UAE—Fatima College of Health Sciences). The study questionnaire was created using *Qualtrics*® and JISC® software for electronic distribution. The participants were recruited via email invitations from their respective institutions. All data collection was

conducted between December 2021 and April 2022. The inclusion criteria for participating in the study were an ability to speak and write in English, being above 18 years of age, and being a pre-registration nursing student.

Ethical approval was obtained from committees at each institution by research and ethical review and approval Committee, Oman College of Health Sciences (OCHS/REC/PROPSA-L-APPROVED/1/2021), Standing Committee for Scientific Research, Jazan University (REC-43/03/047), research and ethical review and approval Committee, Fatima College of Health Sciences (78220), and Northumbria University research ethics committee (35636). A written informed consent statement was presented on the first screen of the survey tool and participants were required to click a button to confirm their consent before they could complete the survey. If the participant chose not to consent, then they were directed to the end of the survey.

### 2.2 Measures

Data on demographic and academic backgrounds were gathered, encompassing information such as age, sex, marital status, year of study, and type of learning (online only, online and attendance, attendance only). Additionally, students were asked about their COVID-19 infection, vaccination status, and participation in clinical placements throughout the pandemic.

Fear level was determined using Fear of COVID-19 Scale (FCV-19S) [16]. This is a self-rated 7-item scale scored on 5-point Likert scales ranging from 1 (strongly disagree) to 5 (strongly agree). Total scores range from 7 to 35, with higher scores indicating higher levels of COVID-19-related fear. A cut-off of 17.5 indicates high fear level [17] and was adopted for this study. In previous research, Cronbach's α values were found to be 0.82, which demonstrated good internal reliability [16]. The FCV-19S has been thoroughly validated as a tool to measure fear of COVID-19 among nursing students from various countries and demonstrated excellent internal consistency [18]. In this study the Cronbach's α was 0.88.

Stress was measured using the Perceived Stress Scale (PSS), a 10 item scale with response on a 4-point scale ranging from 0 (never) to 4 (very often), with overall scores ranging from 0 to 40 [19]. Scores ≥14 indicate the presence of moderate stress [19]. The PSS-10 version is reported to have acceptable internal consistency (α = 0.70) [20]. The PSS has undergone extensive validation in nursing students and consistently demonstrated good internal consistency, with a reported Cronbach's alpha coefficient of 0.93 [21]. In the current study Cronbach's α was 0.81.

The Hospital Anxiety and Depression Scale (HADS). comprises 14 items assessing anxiety (7-items; HADS-A) and depression (7-items; HADS-D) [22]. Each is rated on a 5-point response scale (from 0 to 4). The scores in each subscale are computed by summing the corresponding items (possible scores of 0–21 for each. subscale). A score of 0–7 is considered normal, 8–10 a borderline case, and 11–21 a case exhibiting anxiety or depression [22]. The cut-off value of 8 and above for both HADS-A and HADS-D [23] was adopted for this study. The HADS has demonstrated very good internal consistency among nursing student (Cronbach's α = 0.82) [24]. In the current study, Cronbach's α was 0.63 for the HADS-A and 0.63 for the HADS-D.

The Insomnia Severity Index (ISI) is a seven item self-report questionnaire assessing the nature, severity, and impact of insomnia [25]. A 5-point scale is used to rate each item (0 = no problem; 4 = very severe problem), yielding a total score ranging from 0 to 28. The total score is interpreted as follows: absence of insomnia (0–7); sub-threshold insomnia (8–14); moderate insomnia (15–21); and severe insomnia (22–28). Cut-off scores of 10 have been used to identify possibility of insomnia [25]. Previous research has reported adequate psychometric

properties of the ISI among nursing student (Cronbach's α = .86) [26]. The Cronbach's α of the ISI in this study was 0.71.

## 2.3 Data analysis

The questionnaires were thoroughly reviewed for accuracy and completeness after the data collection process. Participants who decided not to respond to certain questions or left one or more questions unanswered were not included in the valid data set due to missing or incomplete data. Data were entered into and analysed using IBM SPSS Statistics (Version 25.0). Frequencies and percentages were used as descriptive statistics. Caseness for each of the psychometric constructs was assigned dichotomously using the cut-off scores indicated above. The Chi-square test was utilised for comparisons to determine group differences. Pearson's correlation coefficient was used to determine the relationships among variables. Logistic regression was performed, to understand the relationships between high level of fear, stress, anxiety, depression, insomnia, and other study variables. For all analyses performed, a $P < .05$ was considered statistically significant.

## 3 Results

A total of 1053 questionnaires were collected. After excluding incomplete questionnaires, the study was left with 918 valid questionnaires from nursing students. Most of the participants were female (85.7%, n = 787), and single (86.9%, n = 798). The largest age group was those aged 21 to 22 years (47.2%, n = 443). The participants came from a range of academic year of study: 33.7% fourth years, 33.1% third years, 23.9% second years, and 9.4% first years. About 70 percent had engaged in a mixed type of learning. The majority had received two or more doses of the COVID-19 vaccination (96.1%, n = 882). One thirds had been infected by COVID-19 (32.6%, n = 299). The mean (SD) of FCV-19S was 16.86 (SD = 6.14), PSS was 21.87 (SD = 5.64), HADS-A was 10.25 (SD = 3.53), HADS-D 9.11 (SD = 3.53), and ISI 12.97 (SD = 5.17). Table 1 shows the prevalence rates of each scale, as follows: fear 46.2%, stress 91.6%, anxiety 69.1%, depression 59.8%, and insomnia 73.2%, based on the specified cut-off scores for each scale.

Respondents from KSA (52.5%) were more likely to score at or above the cut-off level on the FCV-19S compared with those from other countries, indicating greater proportion of students with fear of COVID-19. The prevalence of anxiety was significantly higher in UAE 80%, UK 79%, and KSA a 71% than in Oman 59%, $\chi 2 (4) = 20.314$, P< 0.001. A similar difference was observed for depression prevalence in KSA 70%, UAE 64%, and UK 63%, compared to in Oman 41%, $\chi 2 (4) = 23.92$, P< 0.001. Participants from UAE and KSA had significantly higher prevalence of insomnia compared to participants from the UK and Oman. Only for stress were there no significant differences between countries.

Pearson correlation coefficients (Table 2) indicated significant relationships between fear, stress, anxiety, depression, and insomnia scores.

### 3.1 Factors predicting fear, stress, anxiety, depression, and insomnia

All variables were selected for entry into the logistic regression models. The models for fear ($F [17, 918] = 59.92$, P< 0.001), stress (F [17, 918] = 41.23, P = 0.001), anxiety (F [17, 918] = 40.90, P< 0.001), depression (F [17, 918] = 86.14, P< 0.001) and insomnia (F [17, 918] = 75.10, P< 0.001) were statistically significant.

Table 3 shows that married status was the strongest predictor of high fear level with married individuals almost three times as likely to score above the cut-off than unmarried ones (OR 2.91 CI,0.99–8.51, $P = 0.05$). Second- and third-year students, those assigned on placement

**Table 1. Demographic characteristic of participants (N = 918).**

|  | Total | | KSA (n = 442) | | OMAN (n = 289) | | UK (n = 97) | | UAE (n = 90) | | |
|---|---|---|---|---|---|---|---|---|---|---|---|
|  | n | % | n | % | n | % | n | % | n | % | p |
| **Gender** |  |  |  |  |  |  |  |  |  |  | .00 |
| Male | 131 | 14.3 | 65 | 14.7 | 61 | 21.1 | 4 | 4.1 | 1 | 1.1 |  |
| Female | 787 | 85.7 | 377 | 85.3 | 228 | 78.9 | 93 | 95.9 | 89 | 98.9 |  |
| **Age** |  |  |  |  |  |  |  |  |  |  | .00 |
| 18–20 | 280 | 30.5 | 78 | 17.6 | 125 | 43.3 | 54 | 55.7 | 23 | 25.6 |  |
| 21–22 | 433 | 47.2 | 239 | 54.1 | 135 | 46.7 | 22 | 22.7 | 37 | 41.1 |  |
| More than 23 | 205 | 22.3 | 125 | 28.3 | 29 | 10 | 21 | 21.6 | 30 | 33.3 |  |
| **Marital Status** |  |  |  |  |  |  |  |  |  |  | .00 |
| Married | 97 | 10.6 | 60 | 13.6 | 12 | 4.2 | 12 | 12.4 | 13 | 14.4 |  |
| Single | 798 | 86.9 | 375 | 84.8 | 276 | 95.5 | 70 | 72.2 | 77 | 85.6 |  |
| Others | 23 | 2.5 | 7 | 1.6 | 1 | 0.3 | 15 | 15.5 | 0 | 0 |  |
| **Academic Year Level** |  |  |  |  |  |  |  |  |  |  | .00 |
| First | 86 | 9.4 | 16 | 3.6 | 50 | 17.3 | 19 | 19.6 | 1 | 1.1 |  |
| Second | 219 | 23.9 | 83 | 18.8 | 89 | 30.8 | 35 | 36.1 | 12 | 13.3 |  |
| Third | 304 | 33.1 | 161 | 36.4 | 73 | 25.3 | 43 | 44.3 | 27 | 30 |  |
| Fourth | 309 | 33.7 | 182 | 41.2 | 77 | 26.6 | 0 | 0 | 50 | 55.6 |  |
| **Type of Learning at present Time** |  |  |  |  |  |  |  |  |  |  | .00 |
| Face-To-Face | 266 | 29 | 98 | 22.2 | 153 | 52.9 | 3 | 3.1 | 12 | 13.3 |  |
| Online | 15 | 1.6 | 5 | 1.1 | 2 | 0.7 | 5 | 5.2 | 3 | 3.3 |  |
| Mixed | 637 | 69.4 | 339 | 76.7 | 134 | 46.4 | 89 | 91.8 | 75 | 83.3 |  |
| **Have you had a Confirmed case of COVID-19** |  |  |  |  |  |  |  |  |  |  | .00 |
| Yes | 299 | 32.6 | 127 | 28.7 | 61 | 21.1 | 59 | 60.8 | 52 | 57.8 |  |
| No | 619 | 67.4 | 315 | 71.3 | 228 | 78.9 | 38 | 39.2 | 38 | 42.2 |  |
| **Have you taken any of COVID-19 vaccine?** |  |  |  |  |  |  |  |  |  |  | .00 |
| Yes, two doses or more | 882 | 96.1 | 437 | 98.9 | 266 | 92 | 92 | 94.8 | 87 | 96.7 |  |
| Yes, one dose | 29 | 3.2 | 2 | 0.5 | 23 | 8 | 2 | 2.1 | 2 | 2.2 |  |
| No | 7 | 0.8 | 3 | 0.7 | 0 | 0 | 3 | 3.1 | 1 | 1.1 |  |
| **Do you assigned to be on placement during the pandemic** |  |  |  |  |  |  |  |  |  |  | .00 |
| Yes | 449 | 48.9 | 146 | 33 | 161 | 55.7 | 73 | 75.3 | 69 | 76.7 |  |
| No | 469 | 51.1 | 296 | 67 | 128 | 44.3 | 24 | 24.7 | 21 | 23.3 |  |
| **FCV-19S** |  |  |  |  |  |  |  |  |  |  | .00 |
| Low Fear (<17.5) | 494 | 53.8 | 210 | 47.5 | 166 | 57.4 | 69 | 71.1 | 49 | 54.4 |  |
| High Fear (≥17.5) | 424 | 46.2 | 232 | 52.5 | 123 | 42.6 | 28 | 28.9 | 41 | 45.6 |  |
| **PSS** |  |  |  |  |  |  |  |  |  |  | .35 |
| Non-Stress (<14) | 77 | 8.4 | 31 | 7 | 26 | 9 | 12 | 12.4 | 8 | 8.9 |  |
| Stress (≥14) | 841 | 91.6 | 411 | 93 | 263 | 91 | 85 | 87.6 | 82 | 91.1 |  |
| **HADS anxiety** |  |  |  |  |  |  |  |  |  |  | .00 |
| No anxiety (HADS(A) <8) | 284 | 30.9 | 128 | 29 | 118 | 40.8 | 20 | 20.6 | 18 | 20 |  |
| Anxiety (HADS(A) ≥8) | 634 | 69.1 | 314 | 71 | 171 | 59.2 | 77 | 79.4 | 72 | 80 |  |
| **HADS depression** |  |  |  |  |  |  |  |  |  |  | .00 |
| No depression (HADS(D) <8) | 369 | 40.2 | 132 | 29.9 | 169 | 58.5 | 36 | 37.1 | 32 | 35.6 |  |
| Depression (HADS(D) ≥8) | 549 | 59.8 | 310 | 70.1 | 120 | 41.5 | 61 | 62.9 | 58 | 64.4 |  |
| **ISI** |  |  |  |  |  |  |  |  |  |  | .00 |
| No insomnia (ISI <10) | 246 | 26.8 | 94 | 21.3 | 84 | 29.1 | 50 | 51.5 | 18 | 20 |  |
| Insomnia (ISI ≥10) | 672 | 73.2 | 348 | 78.7 | 205 | 70.9 | 47 | 48.5 | 72 | 80 |  |

**Table 2. Relationships between fear, stress, anxiety, depression, and insomnia.**

|  | Fear | Stress | Anxiety | Depression | Total ISI |
|---|---|---|---|---|---|
| **Fear** | 1 | .135** | .092** | .132** | .199** |
| **Stress** |  | 1 | .350** | .294** | .327** |
| **Anxiety** |  |  | 1 | .102** | .153** |
| **Depression** |  |  |  | 1 | .289** |
| **Total ISI Score** |  |  |  |  | 1 |

** $p < 0.01$

during the pandemic, and those from the UK had significantly higher levels of Fear of COVID. Being female was the only significant predictor of stress (OR 2.89[CI, 1.55–5.39], $P < 0.001$). Only being a student from Oman appeared to be a predictor of anxiety (OR 0.34[CI, 0.18–0.64], $P < 0.001$).

Four variables predicted depression: a student being in their second academic year (OR 2.37[CI, 1.70–3.83], $P < 0.001$), being female, those between 21 and 22 years of age, and participants from Oman. The last logistic regression model showed that the significant predictors for insomnia in pre-registration nursing students were being in the first academic year (OR 3.03 [CI, 1.41–6.49], $P < 0.001$), second academic year (OR 1.82[CI, 1.08–3.05], $P = 0.02$) and location in Oman or UK.

**Table 3. Logistic regression analyses of factors associated with higher fear, depression, anxiety, stress, sleep disturbance odds ratio (95% CI).**

|  | Higher Fear OR (95% CI) | P | Stress OR (95% CI) | P | Anxiety OR (95% CI) | P | Depression OR (95% CI) | P | Insomnia OR (95% CI) | P |
|---|---|---|---|---|---|---|---|---|---|---|
| **Gender** |  |  |  |  |  |  |  |  |  |  |
| Male | Ref |  | Ref |  | Ref |  | Ref |  | Ref |  |
| Female |  |  | 2.89(1.55–5.39) | .00 |  |  | 1.49(0.99–2.25) | .05 | 0.61(0.37–1.00) | .04 |
| **Age** |  |  |  |  |  |  |  |  |  |  |
| More than 23 | Ref |  | Ref |  | Ref |  | Ref |  | Ref |  |
| 18–20 |  |  |  |  |  |  |  |  |  |  |
| 21–22 |  |  |  |  |  |  | 0.64(0.44–0.95) | .02 |  |  |
| **Marital Status** |  |  |  |  |  |  |  |  |  |  |
| Others | Ref |  | Ref |  | Ref |  | Ref |  | Ref |  |
| Married | 2.91(0.99–8.51) | .05 |  |  |  |  |  |  |  |  |
| Single |  |  |  |  |  |  |  |  |  |  |
| **Academic Year Level** |  |  |  |  |  |  |  |  |  |  |
| Fourth | Ref |  | Ref |  | Ref |  | Ref |  | Ref |  |
| First |  |  |  |  |  |  | 2.07(1.06–4.03) | .03 | 3.03(1.41–6.49) | .00 |
| Second | 2.07(1.31–3.27) | .00 | 2.48(1.07–5.77) | .03 |  |  | 2.37(1.7–3.83) | .00 | 1.82(1.08–3.05) | .02 |
| Third | 1.68(1.17–2.40) | .00 |  |  |  |  |  |  |  |  |
| **Do you assign a placement during the pandemic** |  |  |  |  |  |  |  |  |  |  |
| No | Ref |  | Ref |  | Ref |  | Ref |  | Ref |  |
| Yes | 1.91(1.40–2.60) | .00 | 1.99(1.15–3.44) | .01 |  |  |  |  | 1.51(1.07–2.14) | .02 |
| **Country** |  |  |  |  |  |  |  |  |  |  |
| UAE | Ref |  | Ref |  | Ref |  | Ref |  | Ref |  |
| KSA |  |  |  |  |  |  |  |  |  |  |
| Oman |  |  |  |  | 0.34(0.18–0.64) | .00 | 0.42(0.24–0.74) | .00 | 0.48(0.25–0.93) | .04 |
| UK | 0.37(0.19–0.72) | .00 |  |  |  |  |  |  | 0.13(0.06–0.27) | .00 |

## 4 Discussion

This study was conducted to determine the psychological disorders associated with the COVID-19 pandemic among pre-registration nursing students in four countries. It is one of the few large-scale studies that has assessed the effects of the pandemic on various aspects of psychological health in this group.

The study found that the pre-registration nursing students experienced symptoms of stress from mild to severe (91.6%), anxiety (69%), depression (59.8%), and insomnia (73.2%). Rates in this study were higher compared to those in a systematic review and meta-analyses of 17 studies conducted by Mulyadi et al. [27] involving 13,247 nursing students where prevalence rates of probable stress were (30%), anxiety (32%), depression (52%), and insomnia (27%) during the COVID-19 pandemic. In addition, prevalence rates in this study were higher in comparison with those in the general population in a meta-analyses of 999 studies where rates were: stress (32%) anxiety (28%), depression (27%), and insomnia (32%) during the COVID-19 pandemic [28]. The differences between the findings might be explained by the various isolation measures that were used by the countries to limit the spread of COVID-19. This is important as the timing of the implementation of these measures can influence the severity of the adverse psychological sequelae [29]. Besides the different methods used in these studies, the definitions and sampling methods used also made the comparisons between the findings difficult to interpret.

The mean level of fear of COVID-19 was 16.86 (SD = 6.14) out of a possible score of 35. This finding is in line with the rather limited previous literature [30–32]. Yet, scores in this study were lower than those reported from studies conducted in Mexico (25.71) [33] and Thailand (25.6) [34]. This may be explained by those studies having been conducted at the start of the pandemic when there was greater uncertainty of the effects of COVID-19 on humans. Another possible reason might also be attributed to differences in vaccine availability and distribution, impacting individuals' perception of safety and well-being between the studies.

In this study, there were significant differences between countries in terms of prevalence of fear, anxiety, depression, and insomnia. The reasons for this might be due to variation in individual countries at the time of conducting the study in terms of the level and nature of COVID-19 restrictions including social isolation, lockdown, and the associated economic challenges. Additionally, the variations in mental health outcomes among countries can also be influenced by factors like the availability and accessibility of healthcare services, public health infrastructure, cultural attitudes towards mental health, and the level of social support systems in place during the pandemic [35]. These country-specific elements contribute to the diverse impacts on individuals' mental well-being during the global health crisis.

The results of this study indicate a significant association between stress, anxiety, depression, and insomnia and fear of COVID-19. Previous studies reported similar findings that indicated a direct influence between the stress, anxiety, depression, insomnia and higher level of fear among undergraduate nursing students [36–38]. This suggests that the effects of the pandemic on psychological wellbeing were broad and that a range of strategies might be required to reduce the variety of symptoms in future.

The results suggested that being married is the strongest predictor of a higher level of fear. A possible reason for this is the added contribution of fear of transmitting the infection to their families; this seems especially likely given that more than half of these participants were assigned to clinical placement during the pandemic. In the present study, being female was the strongest predictor of both stress and anxiety. These results are consistent with other studies [39, 40].

Academic year of study was the strongest predictor of depression, namely second year and insomnia, namely first year. This might be due to repercussions on education in terms of

barriers and difficulties of lack of technology skills [41]. This finding suggests that the pandemic may have had a greater impact on those in the early part of the nursing course who would have greater uncertainty about clinical placement and may have had insufficient time to forge close support networks with fellow students. Thus, in future pandemics there may be a need to prioritise supportive interventions for students during pandemics can potentially apply to other situations besides pandemics. The underlying principle is that during any crisis or challenging circumstances, individuals may face increased stress, anxiety, depression, insomnia and disruptions in their lives. This can be particularly true for students who might be dealing with additional pressures related to their education and personal lives, such as financial burdens, family responsibilities, health concerns, or social challenges.

The current study revealed a high prevalence of stress, anxiety, and depression even two years after the onset of the pandemic. A systematic review involving 89 studies found that implementing online mental health consultation was beneficial to reduce depression and anxiety and improve psychological well-being of college students [42], suggesting this might be a way to support nursing students in the future. Higher education institutions should provide online training courses and counselling services to help students overcome their psychological problems. Regretfully, research has shown that nursing programs generally do not offer sufficient training on crisis-coping strategies [43, 44] and it is recommended that the nursing education curriculum is strengthened regarding all aspects of infectious diseases.

A possible limitation of this study is that self-reported questionnaires may not accurately reflect mental health problems and the actual level of fear. Recruitment of participants through online platforms and emails might have led to a lower response rate as some individuals may not regularly check and respond to their emails. Another possible limitation is the large differences in spread of the virus, death rates, and restrictions in different countries.

## 5 Conclusion

This is the first study to examine the psychological impact of COVID-19 pandemic among undergraduate nursing students in KSA, Oman, UK, and UAE. This study revealed that the rate of fear of COVID-19, stress, anxiety, depression, and insomnia among undergraduate nursing students is between moderate to severe level. The main implication of the current findings is that higher education institutions need to support and monitor students' psychological needs and explore the benefits of interventions to reduce the level of psychological symptoms. Likewise, it is important to embed and include disaster and pandemic management in the nursing curriculum to increase the student knowledge and preparedness for such emergencies. Further, researchers should continue to examine the mental health symptoms in this population as the impact of the pandemic may persist over time. Nursing students of today will in the future be working as nurses, and it is important to nurture them at the time of outbreaks of infectious diseases.

## 6 Relevance for clinical practice

Higher education Institutions should consider the various factors that affect the development and maintenance of stress, anxiety, depression, and sleep among undergraduate nursing students. These include their backgrounds, perceptions, and feelings, to minimize their risk of experiencing these conditions in the future. Professional and academic tutors should additionally provide support and help students develop resilience during their internship. In addition to training students how to take care of a patient with an infectious disease, disaster management education should also be included in the nursing curricula to help students develop effective disaster management response skills.

## Acknowledgments

Gratitude to all individuals who took part in the research for their willingness to give up their time to complete the questionnaires.

## Author Contributions

**Conceptualization:** Mohammed Al Maqbali, Norah Madkhali, Geoffrey L. Dickens.

**Data curation:** Mohammed Al Maqbali, Norah Madkhali, Alexander M. Gleason, Geoffrey L. Dickens.

**Formal analysis:** Mohammed Al Maqbali, Norah Madkhali, Alexander M. Gleason, Geoffrey L. Dickens.

**Funding acquisition:** Mohammed Al Maqbali, Norah Madkhali, Alexander M. Gleason, Geoffrey L. Dickens.

**Investigation:** Mohammed Al Maqbali, Norah Madkhali, Alexander M. Gleason, Geoffrey L. Dickens.

**Methodology:** Mohammed Al Maqbali, Norah Madkhali, Alexander M. Gleason, Geoffrey L. Dickens.

**Project administration:** Mohammed Al Maqbali, Norah Madkhali, Geoffrey L. Dickens.

**Resources:** Mohammed Al Maqbali.

**Software:** Mohammed Al Maqbali, Geoffrey L. Dickens.

**Supervision:** Mohammed Al Maqbali, Geoffrey L. Dickens.

**Validation:** Mohammed Al Maqbali.

**Visualization:** Mohammed Al Maqbali.

**Writing – original draft:** Mohammed Al Maqbali, Norah Madkhali, Alexander M. Gleason, Geoffrey L. Dickens.

**Writing – review & editing:** Mohammed Al Maqbali, Norah Madkhali, Alexander M. Gleason, Geoffrey L. Dickens.

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
