## [Decision Letter · Decision Letter 0]

25 Jul 2023

PONE-D-23-04858Fear, Stress, Anxiety, Depression and Insomnia Related to COVID-19 among Undergraduate Nursing Students: An International SurveyPLOS ONE

Dear Dr. Dickens,

Thank you for submitting your manuscript to PLOS ONE. After careful consideration, we feel that it has merit but does not fully meet PLOS ONE’s publication criteria as it currently stands. Therefore, we invite you to submit a revised version of the manuscript that addresses the points raised during the review process.

We look forward to receiving your revised manuscript.

Kind regards,

Ramona Bongelli, Ph.D.

Academic Editor

PLOS ONE

Journal Requirements:

2. For studies reporting research involving human participants, PLOS ONE requires authors to confirm that this specific study was reviewed and approved by an institutional review board (ethics committee) before the study began. Please provide the specific name of the ethics committee/IRB that approved your study, or explain why you did not seek approval in this case.

Reviewers' comments:

Reviewer's Responses to Questions

**Comments to the Author**

1. Is the manuscript technically sound, and do the data support the conclusions?

Reviewer #1: Yes

Reviewer #2: Partly

Reviewer #3: Partly

2. Has the statistical analysis been performed appropriately and rigorously? 

Reviewer #1: Yes

Reviewer #2: No

Reviewer #3: I Don't Know

3. Have the authors made all data underlying the findings in their manuscript fully available?

Reviewer #1: No

Reviewer #2: No

Reviewer #3: No

4. Is the manuscript presented in an intelligible fashion and written in standard English?

Reviewer #1: Yes

Reviewer #2: Yes

Reviewer #3: No

5. Review Comments to the Author

Reviewer #1: Dear Authors

I've enjoyed reading the manuscript. It is well-written and clear.

I have only few minor comments/ suggestions.

- I would made clear from the abstract (and in method section) that background information will be used as independent variables.

- I'm not sure if the regression comprises all the independent variables at once. If yes, please clarify. If no, I would suggest to make regression controlling by country. This would allow to strengthen the generalisability of the findings across countries.

- I would avoid the use of the word "psychological distress" (stress, anxiety, depression) and I would rather use the more general "psychological or psychophysical disease or suffering" given the outcome variables used in the present study.

- I would strengthen discussion (with more sentences about comparisons across the countries both within the study and conducted worldwide after two years of pandemic) and study implications sections (mainly for nursing and healthcare context).

Sincerely

Reviewer #2: Dear Editor,

I am grateful for the opportunity to review the manuscript Fear, Stress, Anxiety, Depression and Insomnia Related to COVID-19 among Undergraduate Nursing Students: An International Survey. The objective of this research was to assess levels of fear, stress, anxiety, depression, and insomnia among undergraduate nursing students in five countries two years after the start of the pandemic. The study sample consisted of 918 undergraduate students who answered the questionnaire online. After reading the manuscript, I forward the following considerations:

1 – The introduction is too generic and the importance of this research is not clear.

2 – The objective variables are not addressed in the introduction, allowing the identification of gaps that will be filled.

3 – The justification for carrying out the research after two years of the pandemic is insufficient to justify the study. It should be noted that there are already many publications on this subject, and it is not noticeable what differentiates this research from the others.

4 – In the methods there is no characterization of the sample.

5 – The survey was carried out via the web and had the premise that students knew the English language. However, there is no way to be sure that this occurred and this may be a bias in this study, since 52.5% of the participants were from Saudi Arabia.

6 – The study used a scale with a Cronbach's alpha value lower than 0.80, which could be a problem for the instrument's internal consistency.

7 – The results are in agreement with the methods.

8 – The discussion shows that the study is similar to several other studies already published. The authors must indicate advances in relation to what is already known.

Given the above, I recommend rejecting the manuscript.

Reviewer #3: Please see specific comments for feedback. Thank you for the opportunity to review this interesting manuscript.Overall, this is a well-designed study and well-organized manuscript. It would benefit from copyediting (see several comments regarding this) and further development of some areas.

Ethics Statement did not address the form of consent obtained

Data availability does not seem to meet the requirements of the publication (available only on request, without further explanation of reason for this).

The manuscript refers to “5 countries” but only four are described: UAE, Saudi Arabia, UK, and Oman. I’m wondering if there might be several countries within the UK that contributed, but this needs further clarification.

p. 3 line 12-13: please rephrase for clarity “many nursing students were needed to undertake clinical practice placements in addition to their education.” Do you mean that healthcare facilities needed the students to complete practice placements or that students needed these placements for their education?

p. 3 lines 14-15: delete second “that” in “It is known that, even outside the context of a global pandemic, that nursing students…”

Additional information about the universities involved would be helpful. Was there just one institution per country? It is implied that all offer a 4-year degree but this is not explicit. It would be helpful to know this information and any other information that might help readers know if these results may be applicable to their settings.

2.1 Study Design: Since students participated in this study it would be helpful to know if they were approached by researchers who were or were not directly involved in their education. Along the same lines, was it made explicit to potential participants that participation in the study would not affect their grades/standing in their program?

p. 4 lines 16-17: “being an pre-registration nursing student”: please change “an” to “a”

p. 4 line 23: “Demographic and academic background data were collected included age, sex…” Please change wording to make this sentence clearer.

p. 5 lines 9-10: If potential responses range from 0-4, wouldn’t this be a 5-point scale?

p. 5 line 11: formatting of reference should be changed (Jahrami et al., 2020)

2.2 Measures: have any of these been used in this population or a similar ones before (students, nurses, nursing students)?

p. 5 line 18: reference format should be changed (Zigmond and Snaith, 1983)

2.2 Measures: please be more explicit about the cutoff scores adopted for this study for the PSS and ISS.

2.3 Data Analysis: Were data screened for missing, implausible, and outlier values?

3 Results: what was the response rate for the survey? How much data were missing in the questionnaires received?

p. 6 line 13: “A total of 918 nursing students participated on the study.” Please change “on” to “in”

p. 6 lines 18-20: It would help the reader if you could present the means (SDs) for each instrument immediately next to the name of the instrument.

p. 6 lines 20-21 “The prevalence of each construct using the identified cut-off score for each of scale” Please re-word for clarity

p. 7 line 17: Please change “iof” to “of”

p. 7 lines 18-19 “Only being a student from Oman appeared to be the most potent predictor of anxiety” Please clarify; was this the only predictor of anxiety or the strongest predictor of anxiety?

p. 7 line 20: please delete “first”

p. 7 line 24: please fix spelling of “adamic”

p. 8 line 9: Mulyadi et al. (2021): please change reference format

p. 8 lines 11-13: Please clarify if this was during the COVID-19 pandemic: “In addition, prevalence rates in this study were higher in comparison with those in the general

12 population in a meta-analyses of 999 studies where rates were: stress (32%) anxiety (28%),

13 depression (27%), and insomnia (32%) [24].”

p. 8 lines 13-14 “The differences between the findings might be explained by the various isolation measures that were used by the countries to limit the spread of COVID-19.” This is one explanation but needs further explanation for those not familiar with isolation measures in the countries studied; were these measures stricter or longer lasting in the countries in this study compared to those included in the reviews you cite?

p. 8 “This is important as the timing of the implementation of these measures can influence the severity of the adverse psychological sequelae.” This statement should be supported by a reference.

p. 8 “This may be explained by those studies having been conducted at the start of the pandemic when there was greater uncertainty of the effects of COVID-19 on humans.” Could this also be explained by the availability of a vaccine at the time of your study compared to vaccine availability in the studies in Thailand and Mexico?

p. 8-9 “The reasons for this might be due to variation in individual countries at the time of conducting the study in terms of the level and nature of COVID-19 restrictions including social isolation, lockdown, and the associated economic challenges.” You may not have an answer to this, but if you do, it may be helpful for your readers to know what sort of country to country differences could explain the significant differences found between countries.

Several times the authors mention data about how many participants were assigned to clinical placement during the pandemic, but it isn’t clear that this data was collected in section 2.2

p. 9 line 14: “Academic year of study was the strongest predictor of depression, namely second year) and…” please revise for clarity; possible remove the parenthesis.

p. 9 lines 19-20: “Thus, in future pandemics there may be a need to prioritise supportive interventions for these students.” Do you think these finding might apply to other situations besides pandemics?

p. 9 line 24: please change “will” to “well”

p. 10 lines 1-2: “Unfortunately, nursing programs not provide sufficient training in how to cope with crises” please revise for clarity

p. 10 lines 5-6: “The participants were recruited via online platform and email, which may have reduced limited the response rate…” please revise for clarity

p. 11 lines 2-3: “training students how to take care of the infection diseases patient” Please consider using “person first” language (for example, “training students how to take care of a patient with an infectious disease”).

Table 1: The abbreviation KSA has not been used elsewhere in the manuscript and needs explanation please

Table 2: In the manuscript it states that “For all analyses performed, a P < .05 was considered statistically significant.” However, this table has flagged results with p <.01.

6. PLOS authors have the option to publish the peer review history of their article (what does this mean?). If published, this will include your full peer review and any attached files.

Reviewer #1: No

Reviewer #2: No

Reviewer #3: No

---

## [Author Response · Author response to Decision Letter 0]

20 Aug 2023

Pleases seen file attached Reply to the Editors and Reviewers, our responses to the reviewers’ comments are described in a point-to-point manner.

---

## [Decision Letter · Decision Letter 1]

13 Sep 2023

PONE-D-23-04858R1Fear, Stress, Anxiety, Depression and Insomnia Related to COVID-19 among Undergraduate Nursing Students: An International SurveyPLOS ONE

Dear Dr. Dickens,

Thank you for submitting your manuscript to PLOS ONE. After careful consideration, we feel that it has merit but does not fully meet PLOS ONE’s publication criteria as it currently stands. Therefore, we invite you to submit a revised version of the manuscript that addresses the points raised during the review process.

We look forward to receiving your revised manuscript.

Kind regards,

Ramona Bongelli, Ph.D.

Academic Editor

PLOS ONE

Journal Requirements:

Reviewers' comments:

Reviewer's Responses to Questions

**Comments to the Author**

1. If the authors have adequately addressed your comments raised in a previous round of review and you feel that this manuscript is now acceptable for publication, you may indicate that here to bypass the “Comments to the Author” section, enter your conflict of interest statement in the “Confidential to Editor” section, and submit your "Accept" recommendation.

Reviewer #3: (No Response)

2. Is the manuscript technically sound, and do the data support the conclusions?

Reviewer #3: Yes

3. Has the statistical analysis been performed appropriately and rigorously? 

Reviewer #3: I Don't Know

4. Have the authors made all data underlying the findings in their manuscript fully available?

Reviewer #3: No

5. Is the manuscript presented in an intelligible fashion and written in standard English?

Reviewer #3: Yes

6. Review Comments to the Author

Reviewer #3: Thank you for your careful attention to the comments provided by the reviewers. I believe that this manuscript has been strengthened by these revisions. I have one minor points that I believe the authors should still address: the PSS is still described as a 4-point scale in the revised manuscript but the authors stated in their response to reviewers table that it was a 5-point scale (p. 6 lines 4-5).

7. PLOS authors have the option to publish the peer review history of their article (what does this mean?). If published, this will include your full peer review and any attached files.

Reviewer #3: No

---

## [Author Response · Author response to Decision Letter 1]

18 Sep 2023

Dear Editor:

Thank you for reviewing our manuscript.

We have amended the manuscript as the reviewers required, addressing their comments below.

Journal Requirements: 

Please review your reference list to ensure that it is complete and correct. If you have cited papers that have been retracted, please include the rationale for doing so in the manuscript text or remove these references and replace them with relevant current references. Any changes to the reference list should be mentioned in the rebuttal letter that accompanies your revised manuscript. If you need to cite a retracted article, indicate the article’s retracted status in the References list and also include a citation and full reference for the retraction notice. 

Author Response: 

Double checked.

Reviewer #3 

Thank you for your careful attention to the comments provided by the reviewers. I believe that this manuscript has been strengthened by these revisions. I have one minor points that I believe the authors should still address: the PSS is still described as a 4-point scale in the revised manuscript, but the authors stated in their response to reviewers table that it was a 5-point scale (p. 6 lines 4-5). 

Author Response: 

Thank you for the comments. Yes, the reviewer is correct, and we have modified the sentence on the text. The range will be as following: 

0= Never 

1= Almost never 

2= Sometimes 

3= Fairly often 

4= Very often

In total will be 5-point scale

The sentences in the manuscript page 6 line 4-5 was modified. 

‘Each is rated on a 5-point response scale (from 0 to 4).’

---

## [Editor Report · Decision Letter 2]

21 Sep 2023

Fear, Stress, Anxiety, Depression and Insomnia Related to COVID-19 among Undergraduate Nursing Students: An International Survey

PONE-D-23-04858R2

Dear Dr. Dickens,

We’re pleased to inform you that your manuscript has been judged scientifically suitable for publication and will be formally accepted for publication once it meets all outstanding technical requirements.

Kind regards,

Ramona Bongelli, Ph.D.

Academic Editor

PLOS ONE
---

## [Editor Report · Acceptance letter]

26 Sep 2023

PONE-D-23-04858R2 

Fear, Stress, Anxiety, Depression and Insomnia Related to COVID-19 among Undergraduate Nursing Students: An International Survey 

Dear Dr. Dickens:

I'm pleased to inform you that your manuscript has been deemed suitable for publication in PLOS ONE. Congratulations! Your manuscript is now with our production department. 

Kind regards, 

on behalf of

Professor Ramona Bongelli 

Academic Editor

PLOS ONE